# Diagnosing Single and Multiple Drug Hypersensitivity in Children: A Tertiary Care Center Retrospective Study

**DOI:** 10.3390/children9121954

**Published:** 2022-12-12

**Authors:** Katarina Milosevic, Marija Malinic, Davor Plavec, Zoran Lekovic, Aleksa Lekovic, Mina Cobeljic, Snezana Rsovac

**Affiliations:** 1Department of Pulmonology and Allergology, University Children’s Hospital, Tirsova 10, 11000 Belgrade, Serbia; 2Faculty of Medicine, University of Belgrade, Dr. Subotica 8, 11000 Belgrade, Serbia; 3Clinic of Dermatovenereology, University Clinical Center of Serbia, Deligradska 34, 11000 Belgrade, Serbia; 4Medical Faculty Osijek, J. J. Strossmayer University of Osijek, Josipa Huttlera 4, 31000 Osijek, Croatia; 5Srebrnjak Children’s Hospital, Srebrnjak 100, 10000 Zagreb, Croatia; 6Department of Gastroenterology, University Children’s Hospital, Tirsova 10, 11000 Belgrade, Serbia; 7Department of Pediatric and Neonatal Intensive Care, University Children’s Hospital, Tirsova 10, 11000 Belgrade, Serbia

**Keywords:** allergy diagnostic algorithm, antibiotics, children, cutaneous manifestations, multiple drug hypersensitivity

## Abstract

Drug hypersensitivity reactions (DHRs) are a type of adverse drug reactions with heterogeneous pathophysiological mechanisms and a broad spectrum of clinical manifestations. Since over-diagnosing is common in children, a complete allergy work-up is needed. A cross-sectional study was conducted at a tertiary care institution, covering the five-year period. Five hundred and four patients of both sexes, mean age 7.5 and with a medical history suggestive of DHR were evaluated. ENDA/EAACI guidelines were used for a diagnostic algorithm. Single drug hypersensitivity was registered in 375 patients and multiple drug hypersensitivity in 129. The main culprits in medical history were antibiotics (83%), non-steroidal anti-inflammatory drugs (NSAIDs) (8.4%) and analgoantipyretics (3.8%). Skin involvement was registered in 96.2%. DHRs were confirmed in 4.4% patients—six patients had positive skin tests and 13 had a positive drug provocation test. In the proven DHRs group, the main agents were antibiotics (72.7%), followed by NSAIDs (8.3%), and of all the skin manifestations, urticaria was most common (78.2%), followed by exanthema (10.5%) and angioedema (5.3%). Considering the above, anticipating DHRs and a proper referral of children to an allergologist is a key step in the assessment of drug hypersensitivity. A complete allergy work-up prevents unnecessary drug exclusion and allows most children to safely continue the use of first-line medications when needed.

## 1. Introduction

The World Health Organization (WHO) defined an adverse drug reaction (ADR) as any response to a drug which is noxious and unintended, and which occurs at doses normally used for the prophylaxis, diagnosis, or therapy of a disease or for the modification of physiological function [1]. Drug hypersensitivity reactions (DHRs) clinically resemble allergic reactions. The term drug allergy can be used if a definite immunological mechanism has been demonstrated [2]. If non-immunological mechanisms are suspected, a drug provocation test (DPT) is essential for making the diagnosis, because in vitro and skin tests (ST) would be negative. Those types of reactions are often caused by non-steroidal anti-inflammatory drugs (NSAIDs) [3]. Multiple studies indicate that among the reported DHRs only a small percentage is confirmed after an allergy work-up [2].

Clinical manifestations usually include skin involvement and manifest as gastrointestinal (nausea and vomiting) and respiratory symptoms (dyspnea) [4].

Easily accessible over-the-counter (OTC) drugs and polypharmacy are the leading cause of multiple drug hypersensitivity (MDH) [3].

The overreporting of DHRs might have consequences and affect treatment management in children labeled with drug allergy. The aim of this study was to determine the prevalence of confirmed cases of DHRs, to analyze suspected and confirmed culprits, as well as clinical manifestations in outpatients managed at our institution.

## 2. Materials and Methods

A cross-sectional study was carried out at the University Children’s Hospital, Belgrade, Serbia, covering the period between 1 January 2016 and 31 December 2020. The study included outpatients of both sexes and all age groups whose medical history referred to a single or MDH. Reactions were divided into immediate and non-immediate reactions based on the time of drug intake and the time when clinical manifestations developed, in accordance with the recommendations of the European Network on Drug Allergy and the European Academy of Allergy and Clinical Immunology (ENDA/EAACI). Immediate reactions occured typically within the first hour after the drug administration, while the onset of non-immediate reactions appeared at least one hour after administration [5]. Patients underwent in vitro testing when possible. Skin tests (prick, intradermal, patch) and DPT represent in vivo tests and were carried out subsequently in terms of negative STs. Implying in vitro and in vivo testing, the ENDA/EACCI diagnostic algorithm was used for the evaluation of suspected DHRs. [5,6] Patients who did not complete a diagnostic algorithm were excluded from the study. Prior to the further evaluation, all parents or guardians were informed about the possible risks of skin and challenge tests, and required to sign a written informed consent. This is retrospective research, in which anonymous patient data from the medical records are used. The study was conducted following the ethical standards of the Ethics Committee of Medical Faculty, University of Belgrade and the Declaration of Helsinki. The approval code provided by the Ethics Committee of Medical Faculty, University of Belgrade is 16225-4510/4-17.

Medical records of interest were collected based on ICD-10 codes that are suggestive of DHRs. Patients with a suspected DHR to antibiotics, NSAIDs, AAs, drugs used for general and local anesthesia, neuromuscular blocking agents (NMBAs) and vaccines were evaluated in this study. The analysis was performed on 839 medical histories, of which 504 patients met all the inclusion criteria. In all, 335 patients were excluded from the study, due to an incomplete diagnostic algorithm—they either failed to present for the follow-up, or provided incomplete and insufficiently reliable data to be included in the study.

In patients with immediate reactions, skin prick tests (SPT) and early reading of intradermal tests (IDT) were performed. Histamine was used as a positive control for SPT and IDT and saline solution as a negative control. Late readings of IDTs were performed after 48 and 72 h. If an infiltrated erythema with a diameter larger than 5 mm appeared, it was considered a positive reaction.

In patients with non-immediate reactions, patch tests (PT) were performed. All reagents were applied to the uninvolved skin of the interscapular region, using acrylate adhesive strips with small plates attached to test allergens (Curatest, Lohmann and Rauscher International GmbH and Co. KG, Rengsdorf, Germany). Readings were made, as recommended by Brockow et al. [7]. DPT was carried out in patients who had no contraindications and had negative results in ST, according to ENDA/EAACI guidelines [8].

Category variables are presented in numbers and percentages. For continuous numerical variables the median (MED (range)) was used. For determining the associations between categorical variables, a chi-square test of independence and the Fisher’s exact test were used. For comparing the variances of two dependent variables, the Man-Whitney U test was used. The probability of *p* < 0.05 was considered statistically significant, and the probability of *p* < 0.01 was considered statistically highly significant.

The datasets will be publicly available during a review or earlier upon request.

## 3. Results

There were 6003 outpatient examinations of which 504 children (8.4%, 95% CI 7.7–9.1%) were referred to the tertiary health care center with a suspected DHR. The study included 254 girls (50.4%) and 250 boys (49.6%). The median age was 7.5 years (0.4–20.5). A hypersensitivity to a single drug was suspected in 375 patients, while MDH was noted in 129 patients (two potential causative agents—108 patients; three potential causative agents—19 patients; four potential causative agents—two patients), making a total of 656 medications and 664 manifestations.

There were 157 immediate (23.9%) and 499 non-immediate suspected reactions (76.1%). Out of all suspected causative drugs, the main confirmed problem in patient medical records included antibiotics (83%), followed by NSAIDs (8.4%) and analgoantipyretics (AAs) (3.8%) (Figure 1). The frequency of clinical manifestations and drugs that are suspected to have caused them are shown in Table 1.

### 3.1. Analysis of Confirmed DHRs

Out of 504 children, only 22 (4.4%, 95% CI 2.8–6.5%) had drug hypersensitivity confirmed. Ten out of those were patients in the multiple drug hypersensitivity (MDH) group, meaning they used two or more drugs. One patient used three, and the rest from the MDH group used two drugs. Among the patients with a proven DHR, 15 (78.9%) had immediate reactions and four patients (21.1%) had non-immediate reactions. No patient had a positive in vitro test. ST was positive in six and DPT in 13 patients. Three patients were automatically considered positive without further evaluation. One had anaphylaxis to ibuprofen and two were using TMP-SMX for which routine testing is not available in our country. In the group of patients with confirmed DHR, the most common causing agents were antibiotics (72.7%), of which cephalosporins were responsible for 10 positive results (45.5%), followed by NSAIDs (9.1%) (Figure 2). The most common manifestation was urticaria (78.2%), followed by exanthema (10.5%) and angioedema (5.3%).

The frequency of suspected DHRs to specific drug classes is shown in Figure 1. Confirmed cases of DHR to a culprit drug are demonstrated in Figure 2.

#### 3.1.1. DHRs to Antibiotics

Two children (0.6%) had a proven DHR to penicillin. One patient had a positive DPT to amoxycillin, which resulted in a non-immediate reaction and manifested as exanthema. As for the group with a suspected reaction to crystalline penicillin, DHR was confirmed by a positive DPT in one patient who had an immediate reaction manifested as urticaria.

In the group where cephalosporins were suspected medications there were 105 reactions (16%). DHR was proven in 10 children (9.5%). ST was positive in one child, while DPT was positive in nine children. Eight patients had an immediate reaction after testing, and two had a non-immediate reaction. After testing, the most common skin reaction was urticaria in five subjects (50%), exanthema in four (40%) and angioedema in one patient (10%). Beside the skin involvement, three children also developed extracutaneous manifestations, which included dyspnea (66.7%) and pruritus (33.3%).

In the Macrolide group, 28 reactions (4.3%) were suspicious. DHR was confirmed in two patients (7.1%) of whom one had a positive DPT and one had a positive ST. Both children developed an immediate reaction after examination. One developed urticaria and another exanthema.

#### 3.1.2. DHRs to Analgoantipyretics (AA)

Considering DHRs to NSAIDs there were 55 suspicious reactions (8.4%), mainly provoked by ibuprofen. Only one patient tested positive to ibuprofen on DPT and developed angioedema, and another one developed anaphylaxis (considered proven DHR by default).

In children who were using other AAs, 25 suspicious reactions (3.8%) were noted. The most frequently suspected agent was acetaminophen. No confirmed DHRs were recorded.

#### 3.1.3. DHRs to General Anesthetics (GA) and Neuromuscular Blocking Agents (NMBA)

Regarding the use of anesthetics and muscle relaxants, eight suspicious reactions were registered (1.2%). ST was positive in three patients (37.5% of all suspected DHRs in this group). One patient tested positive for atropine, another one for fentanyl and the third one for rocuronium bromide. All of them developed urticaria after testing. Sedative and hypnotic, chloral hydrate was also found to be responsible for one positive ST, after which it provoked urticaria. All proven DHRs were immediate reactions.

There was no statistically significant difference in age between children with proven DHR compared to those who tested negative in the allergy work-up (*p* = 0.172). Additionally, no statistically significant difference was found between gender and previously mentioned groups (χ^2^ = 0.225, *p* = 0.635). The presence of extracutaneous manifestations was significantly associated with a positive allergy test (*p* = 0.022, OR 4.62, 95% CI 1.05–15.76%).

## 4. Discussion

Considering that ADRs represent an important public health problem and that drug-allergy is often overreported in the pediatric population, it is of significance that a complete allergy-work up is done and the confirmed DHRs registered [4,9]. The prevalence of self-reported drug hypersensitivity ranges between 2.9% to 16.8%. A couple of studies highlighted it was around 10% in pediatric patients [2,9,10,11]. The systemic review of studies analyzing ADRs in outpatient pediatric population reported that the incidence of ADRs ranges between 0.7% and 2.7% [12]. In our study, the frequency of suspected DHRs was 8.4%. These data indicate that more rigorous and meticulous evaluation must be done in GP practice. Sometimes, diagnosing a DHR can be challenging because polypharmacy and concomitant viral infections make it impossible to discern the real cause of certain manifestations. The creation and implementation of national guidelines, which would indicate when to refer a child to an allergologist, would significantly facilitate everyday clinical practice in primary health care centers.

The reported prevalence of confirmed DHRs ranges between 6–9%. [13,14]. In our study, only 22 children had a confirmed DHR (4.4%). In the Republic of Serbia, there are about 1.43 million people younger than 20 years of age [15]. Initial treatment usually occurs at the primary health care centers. Most of the children who are referred for further allergy diagnostics are managed at our hospital, so we can only presume that a true proportion of those who are unnecessarily deprived of proper therapy due to the pseudoDHR, is quite significant. More than 95% of our patients were falsely assumed to have had a DHR. Diagnostics based only on patients’ medical history may affect treatment and lead to the use of even less effective or more expensive drugs. We highlight that we did not have a single positive in vitro test; hence a complete allergy work-up must be mandatory when a DHR is suspected.

In our study, a quarter of patients had a medical history that was suggestive of MDH (25.6%). Those results are in accordance with the reported ranges (11–40%) [16,17]. Goldberg et al. highlight that the risk of developing ADR for patients taking seven or more drugs is as high as 82% [18].

Other studies report a high prevalence of DHR in children younger than 10 years of age. [10,17,19]. In our study, the median for age was 7.5 years. Age extremes in the population are considered to be a risk factor for developing ADRs, but only five infants were reported with a suspected DHR in our study [20].

Multiple studies indicate that β-lactam antibiotics and NSAIDs are the main problems in patient medical history [10,13,14]. Similar data were noted in our research, where the most frequently suspected drugs were, as well antibiotics, NSAIDs and AAs. Antibiotics were held responsible for 83% of suspected DHRs. Over three quarters (77.4%) of assumed hypersensitivity was due to β-lactam antibiotics, semi-synthetic penicillin (52.4%). These high frequencies can be explained by the fact that these antibiotics are commonly prescribed in the pediatric population. Despite many reports suggesting allergy to penicillin, only a small percentage of children should avoid the use of these antibiotics [21,22]. The results in our study corroborate these findings. We have confirmed β-lactam allergy in 2.4% of cases, in which 0.6% (only two patients) was due to penicillin, and the other part of the confirmed reaction was caused by cephalosporins. The latter were proven to be responsible for most of the confirmed cases of DHR (10 patients: 45.5%). Antibiotic allergy labeling may cause several problems. It has been calculated that prescription costs are 30% to 40% higher in patients with suspected penicillin allergy. Additionally, it has been shown that patients with suspected penicillin allergy who use alternative antibiotics, such as fluoroquinolones, clindamycin and vancomycin are at greater risk of developing *Clostridium difficile*, methicillin-resistant *Staphylococcus aureus* and vancomycin-resistant *Enterococcus species* infections, in comparison with controls [21].

NSAIDs and AAs are commonly prescribed for pain and fever management in children. However, a true hypersensitivity to NSAIDs and acetaminophen is scarce [23]. Our results are in line with these data. No patient evidenced adverse effects after using acetaminophen. It can be safely used in children with kidney impairment, and if used properly it should not cause any liver damage. Taking into account the above-mentioned, clinicians might consider acetaminophen as a first-choice drug to relieve symptoms whenever possible.

Hypersensitivity reactions occurring during anesthesia can be fatal and determining the main causing agent can be challenging. Characteristic symptoms such as vertigo, dyspnea and malaise are missing in anesthetized patients. Cutaneous manifestations, such as urticaria and angioedema, sudden cardiac arrest, hypotension, and severe bronchoconstriction are highly evocative of anaphylaxis [24]. Drug hypersensitivity in our patients was confirmed in three children who developed only skin manifestations.

Vaccines and their components, stabilizers, preservatives, and adjuvants, can cause a hypersensitive reaction. However, a true hypersensitivity to vaccines is rare and signs and symptoms appearing after vaccination are coincidental [25]. No vaccine allergy was proven and children with a suspected DHR were clinically processed and safely vaccinated in our allergy department. Vaccination is an indispensable public health action, and it is necessary to evaluate children through an allergy work-up. The WHO has listed vaccine hesitancy as one of the biggest threats to the global health and emphasized the importance of a clinician’s role in promoting vaccine safety.

ADRs can have a broad spectrum of manifestations. Erkoçoğlu et al. found that more than 90% of children had cutaneous manifestations, followed by gastrointestinal and respiratory symptoms [26]. Exanthema and urticaria are often seen as the most frequent skin eruptions in patients with suspected DHRs. A high frequency of skin involvement was present in our study (96.2%) (Table 1). In both groups of children, including those with suspected and proven DHRs, urticaria was the most common skin eruption, followed by exanthema. It is important to emphasize that patients in our research with positive allergy testing significantly more often reported extracutaneous manifestations.

This study has important limitations. It is an observational, retrospective study. The data about the culprit drug, the onset of manifestation and their characteristics were collected by the revision of patients’ medical history, which was based on anamnestic data. This could have had a strong impact on the proper classification of cutaneous manifestations. A multicentric study would more precisely reveal the real number of false DHRs and could be of great significance in everyday clinical practice considering the prescription of medications, especially antibiotics.

The other limitation is linked to cross-reactivity. In all, 22 positive patients had proven DHRs for medications that belong to different drug groups. Having in mind that those drug groups are very heterogenous, and that the sample would have been very small, and frequency very low, we could not conduct statistical analyses and prove or exclude the association between drugs and cross-reactivity.

## 5. Conclusions

To minimize over- or underdiagnosing drug allergies, all children with suspected DHRs should be completely diagnostically evaluated by an allergologist. Additionally, there is a need to make guidelines for pediatricians working in primary health care facilities that would clearly indicate children who need a full allergy work-up. Considering that ADRs recorded in this study were caused by medications commonly prescribed in everyday clinical practice, it is necessary to promote cautious use. Rational and effective pharmacotherapy, as well as the reporting of ADRs, is the key for increasing medication safety.

## Figures and Tables

**Figure 1 children-09-01954-f001:**
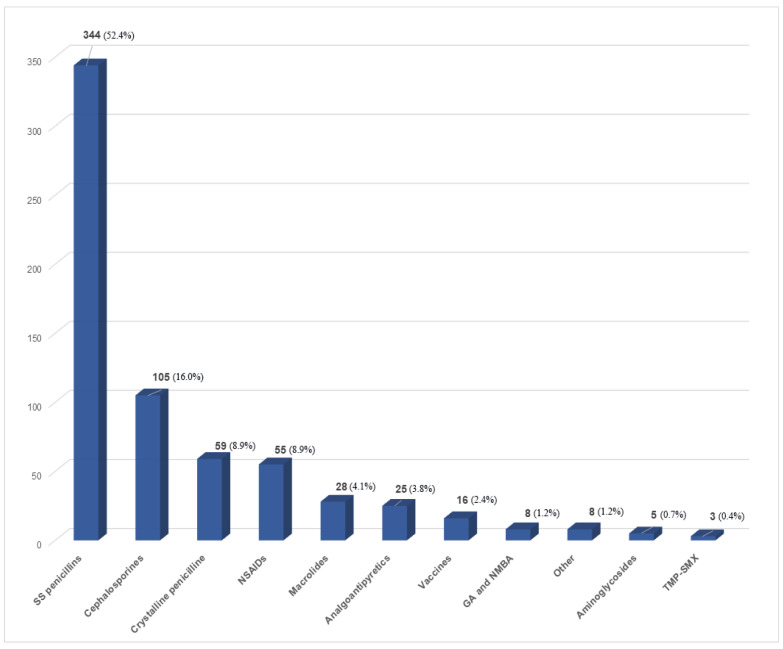
Suspected DHRs to specific drug classes (data obtained from patients’ medical histories). DHRs—Drug hypersensitivity reactions; SS penicillins—semi-synthetic penicillins; NSAIDs—non-steroidal anti-inflammatory drugs; GA—general anesthetics; NMBA—neuromuscular blocking agents; TMP-SMX—trimethoprim-sulfamethoxazole.

**Figure 2 children-09-01954-f002:**
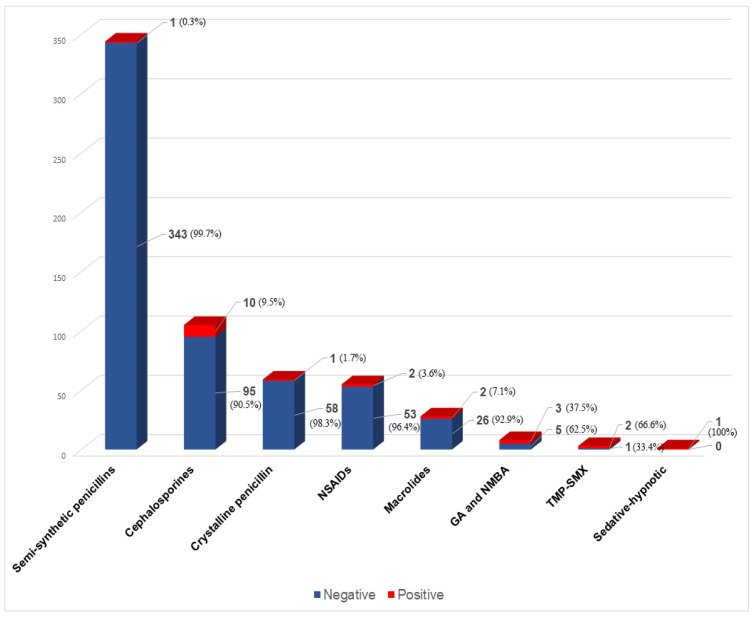
Ratio between proven and excluded DHRs. DHRs—Drug hypersensitivity reactions; SS penicillins—semi-synthetic penicillins; NSAIDs—non-steroidal anti-inflammatory drugs; GA—general anesthetics; NMBA—neuromuscular blocking agents; TMP-SMX—trimethoprim-sulfamethoxazole.

**Table 1 children-09-01954-t001:** Cutaneous and extracutaneous manifestations provoked by suspected drug class (data obtained from patients’ medical histories).

Drug Class	Suspected Adverse Reactions							
	Cutaneous (C)			Total C Adrs within Group	Extracutaneous (EC)			Total EC ADRs within Group
	Urticaria (%) *	Exanthema (%) *	Angioedema (%) *		Dyspnea (%) *	Syncope (%) *	Other **** (%) *	
Aminoglycosides	2 (40)	2 (40)	1 (20)	5	0	0	0	
Analgoantipyretics	14 (56)	5 (20)	6 (24)	25	0	0	0	
Cephalosporines	52 (50)	43 (41.3)	9 (8.7)	104	2 (50)	0	2 (50)	4
Crystalline penicillins	21 (38.9)	29 (53.7)	4 (7.4)	54	0	3 (60)	2 (40)	5
Macrolides	13 (46.4)	11 (39.3)	4 (14.3)	28	0	0	0	
NSAIDs	24 (44.4)	11 (20.4)	19 (35.2)	54	1 *** (50)	0	1 (50)	2
GA and NMBA	1 (20)	2 (40)	2 (40)	5	3 (100)	0	0	3
Other	3 (37.5)	3 (37.5)	2 (25)	8	0	0	0	
Semi-synthetic penicillins	165 (48)	151 (43.9)	28 (8.1)	344	2 (66.7)	1 (33.3)	0	3
TMP/SMX	2 (66.7)	1 (33.3)	0	3	0	0	0	
Vaccines	4 (44.4)	2 (22.2)	3 (33.3)	9	0	0	8 (100)	8
Total ADRs **	301 (45.3)	260 (39.2)	78 (11.7)	639 (96.2)	8 (1.2)	4 (0.6)	13 (1.96)	25 (3.8)

* Percent within the drug class group; ** Percent based on total number of cutaneous and extracutaneous manifestations (*n* = 664); *** Patient developed anaphylaxis; **** Other extracutaneous manifestations were pruritus (1) and rhinoconjunctivitis (1) for Cephalosporines, vertigo (1) and vomiting (1) for Crystalline penicillins, tachycardia (1) for NSAIDs, and in vaccines it was fever (1), somnolence (1), vomiting (1) and five cases of non-stop crying (for more than 3 h) infant; ADR—Adverse drug reactions; GA—General anesthesia; NMBA—Neuromuscular blocking agents; TMP/SMX—Trimethoprim/sulfamethoxazole.

## Data Availability

The datasets will be publicly available during review or earlier upon request.

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
