# Peer review of "Diagnosing Single and Multiple Drug Hypersensitivity in Children: A Tertiary Care Center Retrospective Study"

_children, 2022, doi:10.3390/children9121954_

Round 1

Reviewer 1 Report

DHRs are confirmed in 4.4% of patients (many false positivities) But 335 were excluded for incomplete diagnostic algorithm. (many false negative may exist)

This must be taken in account in the final results of study

Reviewer 2 Report

Review on Milosevic et al’s “Diagnosing single and multiple drug hypersensitivity in children: tertiary health care center experience”

Dear Authors,

1. Please, modify your title according to STROBE checklist (on case control studies)! Indicate the study design with a commonly used term in the title: retrospective. Cross-sectional study is already written in the abstract.  

2. Please, recheck abbreviation usage. E.g: I could not find EA. I suppose it is angioedema, isn’t it?

3. Do you mean “in-an-hour” reactions as immediate ones? 

4. Please, include ethical statement number into “materials and methods” or to “Institutional Review Board Statement”. 

5. Due to incomplete diagnostic algorithm 335 patients were excluded from the study. That is fine, as they have not finished the whole diagnostic process. But can you provide data on the symptoms they had and made the indication for further examination?

6. In Table 1. manifestations provoked by suspected drugs are shown. In case of extracutaneous symptoms what types of symptoms under the word “others” are included? Please, specify in text or in explanation of the certain table. 

7. Microbiological names must be put in italics. (Clostridium difficile and others)

8. Please, revise the word “culprit”! It is alright, but repeated so so many times. If possible at least replace some of it with synonymous words (agent, problem..e.t.c.) according to text environment.

9. Can the authors also answer the following question based on their data: were there more frequent associations between drugs? Did they occur more commonly together? ..or there is no such phenomenon detected in this population. 
